# The Mimic Enzyme Properties of Au@PtNRs and the Detection for Ascorbic Acid Based on Their Catalytic Properties

**Hao Gan** [1,2]**, Wenzhao Han** [1]**, Jiadi Liu** [3]**, Juntian Qi** [1]**, Hui Li** [1] **and Liping Wang** [1,*]

[1]   Key Laboratory for Molecular Enzymology and Engineering, Ministry of Education, School of Life Sciences, Engineering Laboratory for AIDS Vaccine, Jilin Universtiy, Changchun 130012, China; ganhao16@mails.jlu.edu.cn (H.G.); hanwz18@mails.jlu.edu.cn (W.H.); qijt1318@mails.jlu.edu.cn (J.Q.); Lihui18@mails.jlu.edu.cn (H.L.)
[2]   Special Education College, Changchun University, Changchun 130022, China
[3]   School of Plant Sciences, Jilin University, Changchun 130012, China; Liujd8218@mails.jlu.edu.cn
[*]   Correspondence: wanglp@jlu.edu.cn; Tel.: +86-431-8515-5348

**Abstract:** Being superior to natural enzymes, nanoenzymes are drawing a great deal of attention in the field of biosensing. Herein, we developed an ultrasensitive, stable and selective colorimetric assay having dual functionalities of Au-tipped Pt nanorods (NRs). The optical and catalytic properties of Au-tipped Pt NRs were monitored using a spectrophotometer and the chromogenic substrate 3, 3′, 5, 5′-tetramethylbenzidine (TMB) in the presence of $H_2O_2$, respectively. We found that Au-tipped Pt NRs exhibited excellent peroxidase-like activity, which decomposed hydrogen peroxide ($H_2O_2$) into oxygen ($O_2$). The produced $O_2$ oxidized the chromogenic substrate into a blue color product. The oxidation rate of the chromogenic substrate could be monitored using a spectrophotometer at 652 nm. Notably, the peroxidase-like activity of Au-tipped Pt NRs decreased in the presence of ascorbic acid (AA). The produced $O_2$ preferentially reacted with AA, generating ascorbyl radicals (AA·) instead of oxidizing TMB, and thereby decreased the oxidation rate of TMB. Based on this inhibitory property, a selective colorimetric assay was developed using Au-tipped Pt NRs for the detection of AA. This work offers a novel detection method for AA.

**Keywords:** Au-tipped Pt; nanoenzyme; electron transfer; catalytic; detection

## 1. Introduction

Lately, due to its cost-effectiveness, colorimetric assay technology has become remarkably popular in the arena of colorimetric biosensing [1,2]. In addition, the test results obtained from the colorimetric assay technology can be easily analyzed by the naked eye or by using a simple portable device [3–5]. Particularly, these advantages have turned the colorimetric assay technology into an extremely desirable tool for nursing work and field studies [6,7].

Owing to their superior physicochemical properties, Au NRs have attracted immense attention in the diagnostic field [8,9]. Due to the difference in aspect ratio, the longitudinal surface plasmon resonance (LSPR) wavelength of Au NRs is controlled from 650 nm to near-infrared 1000 nm [9,10]. Since Au NRs have catalytic properties, these can also be used in colorimetric assays [11–15]. However, it is a challenging task due to their extremely low sensitivity for this purpose [16]. Thus, it is desirable to explore novel materials that could improve the sensitivity of colorimetric assay [17,18]. Pt nanoparticles have been widely used as a catalyst in several important chemical reactions [19,20]. Thus, Au and Pt nanocomposites, with strong catalytic properties, can be designed as nanoenzymes. In addition, using

these, the sensitivity of colorimetric assay technologies in the colorimetric biosensing field may be greatly improved [21–23].

Ascorbic acid (AA), the main biological form of vitamin C, participates in numerous metabolic processes in the organisms [24]. Therefore, the detection of AA is important for studying various pathophysiological conditions. Popular detection methods for AA include electrochemistry, spectrophotometry, chromatography and the colorimetric method [25,26]. Since the colorimetric method is advantageous due to ease, speed and cost-effectiveness, using a nanoenzyme sensor to detect AA has attracted great attention [27].

Several research groups have studied the nanoenzyme property of Au-tipped Pt NRs and their application as biosensors. Interestingly, the peroxidase activity of Au-tipped Pt NRs has been confirmed with 3,3′,5,5′-tetramethylbenzidine (TMB) [28]. Based on this, using the antioxidant properties, we developed an ultrasensitive and rapid colorimetric assay for AA. Our work explored novel Au-tipped Pt NRs as an enzyme mimic for applications in the biosensing field.

## 2. Results and Discussion

### 2.1. Structure and Morphology Analysis of Au-Tipped Pt NRs

Figure 1a shows a transmission electron microscopy (TEM) image of pure Au NRs having an average length diameter ratio of 3.3. In addition, the Au NRs surface is smooth. Similarly, Figure 1b shows the TEM image of Au-tipped Pt NRs. The Pt nanoparticles are mainly distributed at the tip of the Au NRs. This is because of the Ag ions and cetyltrimethylammonium bromide (CTAB) on the side of the Au NRs, which resulted in a large amount of Pt deposition at the tip (Figure 1c).

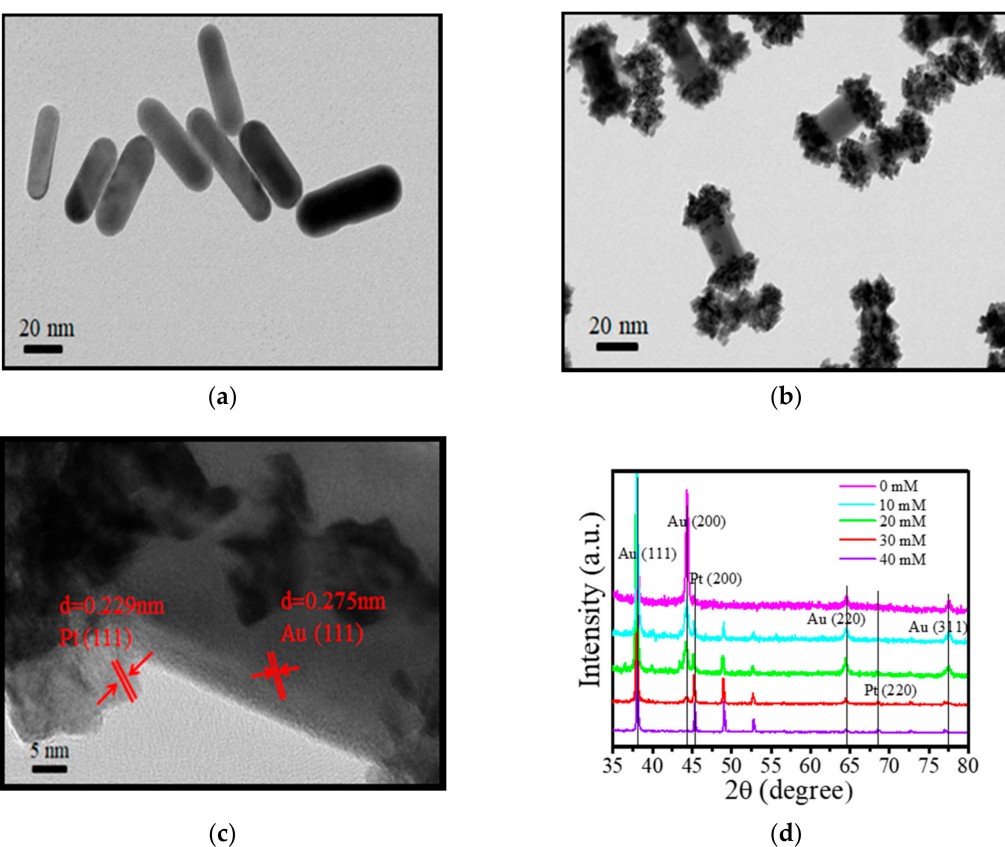

**Figure 1.** TEM images of (**a**) Au nanorods (NRs) and (**b**) Au-tipped Pt NRs. (**c**) HRTEM images of Au-tipped Pt NRs. (**d**) XRD of Au NRs and Au-tipped Pt NRs. The concentrations of Pt were 10, 20, 30 and 40 mM.

The corresponding XRD pattern is shown in Figure 1d. It exhibits sharp diffraction peaks, including (111) (200) (220) and (311) planes of Au NRs. The characteristic absorbance peak of (111) plane is dominantly present in the XRD pattern, suggesting it to be one of the main exposed surfaces. Upon addition of 10 mM Pt to form Au-tipped Pt NRs, a small peak of Pt (200) plane appeared in the XRD pattern (Figure 1d). However, the peak of the Au (200) plane was lower than the Au NRs. Upon further increasing the concentration of Pt ions to 30 mM, another small peak of the Pt (220) plane appeared. Interestingly, simultaneously, the peak of Au (220) and (311) planes reduced significantly. Moreover, when the concentration of Pt was further increased to 40 mM, the peak of the Pt (220) plane remained unchanged. However, the peak of the Pt (200) plane increased. Based on these results, we concluded that Pt nanoparticles interacted with the high lattice energy of Au NRs to form Au-tipped Pt NRs, suggesting that the growth of Pt nanoparticles was on the (200) (220) and (311) planes of Au NRs.

Next, we studied the optical properties of Au-tipped Pt NRs. Figure 2 shows the absorption spectra of Au NRs and Au-tipped Pt NRs. Here, Au NRs exhibited a weak transverse localized surface plasmon resonance (TSPR) band at 510 nm and a strong longitudinal localized surface plasmon resonance (LSPR) band at 724 nm, depositing Pt on Au NRs to form Au-tipped Pt NRs. Upon increasing the concentration of Pt ions, the longitudinal localized surface plasmon resonance bands showed a red-shift from 724 nm to 900 nm. However, the transverse localized surface plasmon band showed no obvious change. This could be due to LSPR having higher polarizability than TSPR. The observed red-shift (176 nm) should have come from the Pt nanoparticles tipped on the Au NRs, due to the change of relative dielectric coefficient. The electrolyte around pure Au NRs was ultrapure water, and the dielectric coefficient of ultrapure water is 78.36 F/m. After the formation of Au-tipped Pt NRs, the electrolyte at the tip of Au NRs changed from water to Pt nanoparticles, which have a much higher dielectric coefficient compared to water. Hence, the LSPR of Au-tipped Pt NRs showed a red-shift compared to Au NRs.

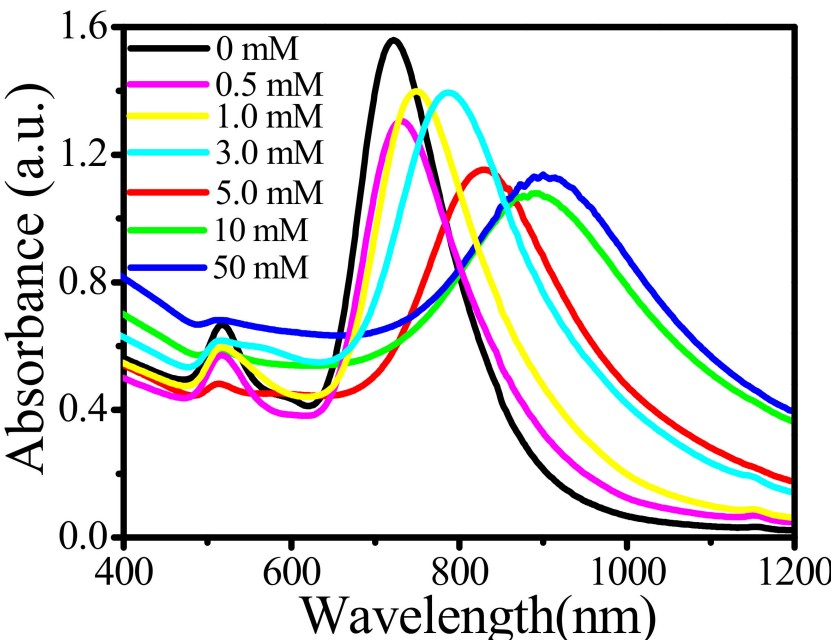

**Figure 2.** Absorption spectra of the Au NRs and the Au-tipped Pt NRs with concentrations of Pt ions of 0, 0.5, 1, 3, 5, 10 and 50 mM.

For Au NRs, two absorption bands, including longitudinal localized surface plasmon resonance (LSPR) at 724 nm and transverse localized surface plasmon resonance (TSPR) at 510 nm, can be seen in the UV-vis absorption spectrum. They correspond to the electron oscillations along the long axis and short axis of Au NRs, respectively. L/D means the length-to-width ratio of Au NRs. The increase of L/D

may induce the red-shift of the LSPR peak. Herein, the L/D of Au NRs remains the same during the preparation of Au-tipped Pt NRs. The optical properties of Au-tipped Pt NRs were studied, as shown in Figure 2. With increasing concentration of Pt ions, the LSPR peak of Au NRs exhibited a red-shift from 724 nm to 900 nm, but the TSPR peak had no obvious change owing to the higher polarizability of LSPR peaks by deposition of Pt nanoparticles on Au NRs. The electrolyte around pure Au NRs was ultrapure water, and the dielectric coefficient of ultrapure water was 78.36 F/m. After the formation of Au-tipped Pt NRs, the electrolyte at the tip of Au NRs was changed from water to Pt nanoparticles. At the same time, the dielectric coefficient was increased because the dielectric coefficient of metal was extremely large. Hence, the LSPR peaks of Au-tipped Pt NRs appeared to red-shift compared with Au NRs.

## 2.2. Peroxidase Activity of Au-Tipped Pt NRs

Herein, Au NRs, Pt NPs and Au@Pt exhibit catalytic properties. The spin-trapping technique was used to confirm the catalytic activity of Au NRs, Pt NPs and Au@Pt NRs, as shown in Figure 3a. The results show that the hydroxyl radical is formed. However, their catalytic properties indicate that Au@Pt NRs are highest among these materials. Hence, Au@Pt NRs were confirmed to own obvious high peroxidase-like activity. Peroxidase activity of Au-tipped Pt NRs was determined using the Langmuir-Hinshelwood mechanism, and the corresponding rate constants were obtained. The oxidation process of TMB is shown in Figure 3a. Results show that the absorption intensity of the reaction mixture increased with time and the catalytic reaction followed pseudo-first-order kinetics as described by the following rate equations:

$$r = -\ln \frac{[oxTMB]}{[oxTMB]_0} = -\ln\left(\frac{I}{I_0}\right) = k_{obs}t \tag{1}$$

$$k_{obs} = -\frac{k(S_0)^2 K_{TMB}[TMB]}{(1 + K_{TMB}[TMB]_0)_2} \tag{2}$$

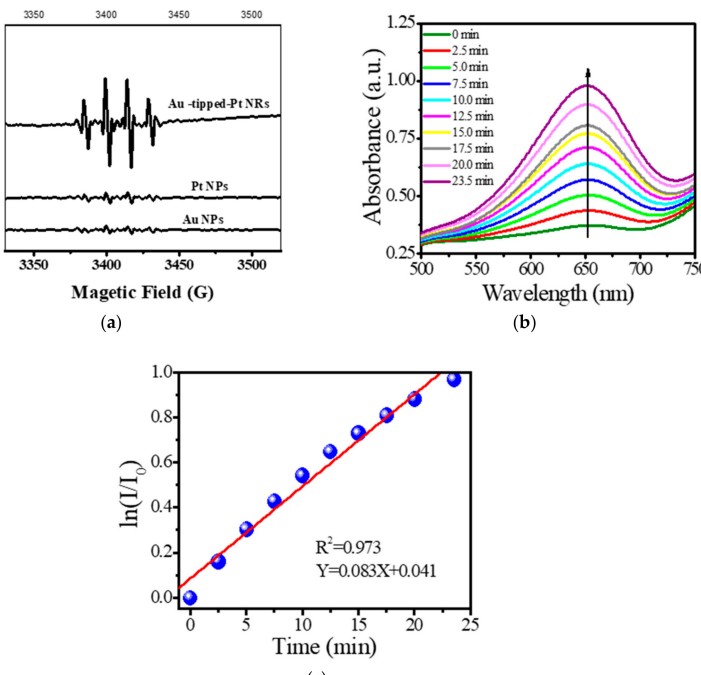

**Figure 3.** (**a**) The spin-trapping spectra of Au NRs, Pt NPs and Au@Pt NRs; (**b**) time-dependent accelerating effect of Au-tipped Pt NRs catalyzed TMB; (**c**) Lineweaver-Burk plots of Au-tipped Pt NRs.

Equations (1) and (2) are the derivative and integral forms of the rate law for the pseudo-first-order reaction, where r is the reaction rate for the catalytic process, $k_{obs}$ is the apparent pseudo-first-order rate constant, $I_0$ and I are the corresponding absorbance of oxTMB (oxidized 3,3′,5,5′-tetramethylbenzidine) at 650 nm at time 0 and at the time point of the measurement. The linear relationship between ln ($I/I_0$) and reaction time (t) is shown in Figure 3b. The linear relationship indicates that the catalytic reaction obeys pseudo-first-order kinetics, and the slope of the linear curve represents the catalytic reaction rate constant ($k_{obs}$), having the value 0.083 min$^{-1}$.

The catalytic mechanism of the reaction is shown in Figure 4. The Fermi level of Au is higher than the Fermi level of Pt, so the work function of Au (5.1 eV) is lower than that of Pt (5.6 eV). Due to the plasma coupling, the electrons on the surface of Au jumped to the hot electron energy level and, being unstable, continually combined with the reactant [29]. Upon prolonging the chemical bonds of the reactants, the bond energy gets reduced, leading to the breaking of chemical bonds. Consequently, the speed of $H_2O_2$ decomposition into $O_2$ gets accelerated, and the rate of TMB oxidation increases. Lastly, the hot electrons get transferred to the platinum surface.

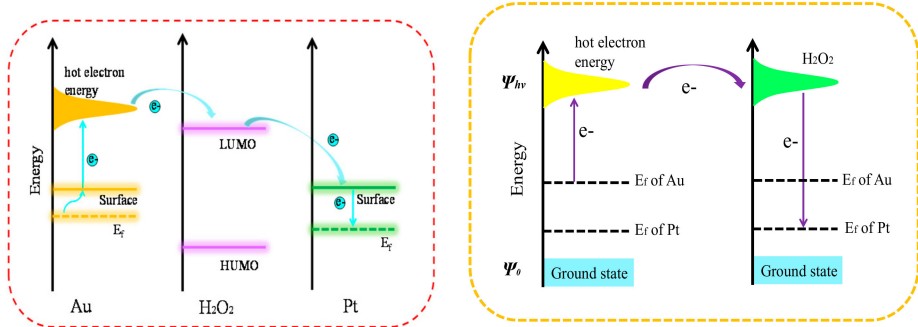

**Figure 4.** Schematic of work-function-driven electron transfer from the surface of Au to the surface of Pt.

## 2.3. The Inhibitory Effects of AA on the Peroxidase Activity

As shown in Figure 5, pure TMB and $H_2O_2$ solutions did not show a noticeable absorption peak at 650 nm. However, upon the addition of Au-tipped Pt NRs, absorption at 650 nm was enhanced, suggesting that the peroxidase property indeed belongs to Au. Moreover, the rate of reaction also increased upon the addition of Au-tipped Pt NRs. When AA was added in the above reaction system, as shown in Figure 5c, the characteristic peak of oxTMB at 650 nm decreased, indicating that AA inhibited the oxidation of TMB.

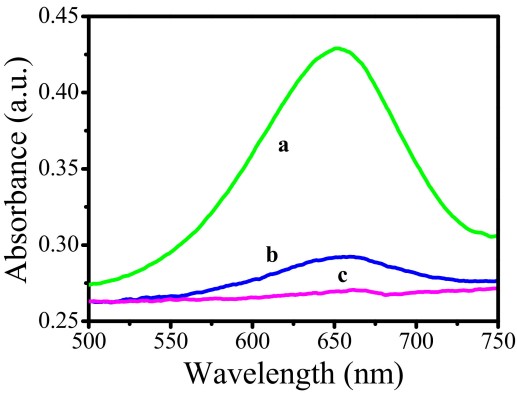

**Figure 5.** Absorption spectra of TMB system in the absence of L-ascorbic acid (AA) and presence of AA. (**a**) 0.8 mL 5, 5′-tetramethylbenzidine (TMB) + 0.2 mL Au-tipped Pt NRs + 0.3 mL $H_2O_2$ for 5 min, (**b**) 0.8 mL TMB +0.1 mL Au-tipped Pt NRs+ 0.3 mL $H_2O_2$ for 5 min, (**c**) 0.8 mL TMB +0.2 mL Au-tipped Pt NRs+ 0.3 mL $H_2O_2$ + 0.2 mL AA for 5 min. The concentration of AA was 1 μM.

The inhibitory effect of AA on the TMB oxidation process is shown in Figure 6. When AA was not added, Au-tipped Pt NRs accelerated the decomposition of $H_2O_2$ to produce $O_2$ at RT and oxidized the substrate molecule TMB to form oxTMB. With an increase of $O_2$ production, correspondingly, more TMB was oxidized. Consequently, the color of the solution also turned from colorless to blue. On the contrary, if AA was present, the oxygen generated from the decomposition of $H_2O_2$ by Au-tipped Pt NRs was consumed by AA instead of oxidizing the TMB (Figure 6). Thereby, no color change of the solution was observed.

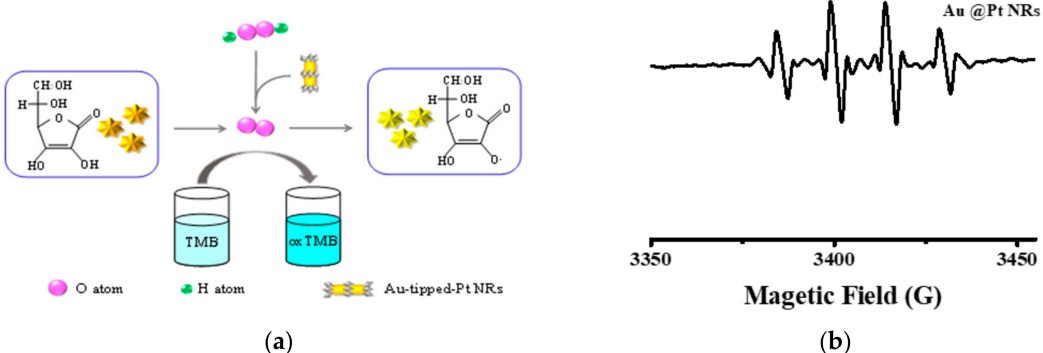

(**a**)          (**b**)

**Figure 6.** (**a**) Schematic illustration of AA as preferential oxidation of the oxidation process of TMB. (**b**) ESR spectra of •OH radicals in the presence of DMPO as a spin trap in the Au NRs/DMPO/H2O2.

The inhibitory effect of AA on the TMB oxidation process is shown in Figure 6a. When AA was not added, Au-tipped Pt NRs accelerated the decomposition of H2O2 to produce •OH at room temperature, which could oxidize the substrate molecule TMB to form oxidized 3,3′,5,5′-tetramethylbenzidine (oxTMB). More oxTMB was produced with more •OH. Figure 6b shows that Au@Pt NRs produced more •OH to oxidize TMB. Meanwhile, the color of the solution turned from colorless to blue. When AA was added, the oxygen generated by Au-tipped Pt NRs decomposition H2O2 first combined with AA to form AA.

## 2.4. Detection of AA by Au-Tipped Pt NRs

Based on the good catalytic performance of Au-tipped Pt NRs, they can be used for the detection of AA. The kinetics of the absorbance change (at 652 nm), upon oxidation of TMB, of the system of Au-tipped Pt NRs–TMB with the addition of different concentrations of AA is shown in Figure 7a. The concentration of AA and absorption intensity of oxTMB follows a linear relationship (Figure 7b). The absorbance of oxTMB linearly increased with the increasing concentration of AA. Therefore, AA can be detected by measuring the oxidation of TMB catalyzed by Au-tipped Pt NRs. In this test, for AA, the limit of detection (LOD) was 0.03 μM. In addition, compared with other detection methods, this colorimetric method is more convenient [22,29].

To realize the specificity of TMB–Au-tipped Pt NRs–$H_2O_2$ system for AA detection, the absorption intensity of 650 nm was studied at different interferences, as shown in Figure 8. The total volume of the detection system was 1.5 mL. The concentration of AA was kept at 1 μM, and the concentration of interferent was also diluted with water up to 1 μM. Then, 0.8 mL TMB and 0.2 mL Au-tipped Pt NRs were mixed with 0.3 mL $H_2O_2$. Next, 0.2 mL of interferential substances was added to the reaction solution, and the reaction mixture was incubated at RT for 5 min. Finally, the absorbance at 650 nm was recorded using a spectrophotometer. Overall, results demonstrated good selectivity for AA detection in this colorimetric assay.

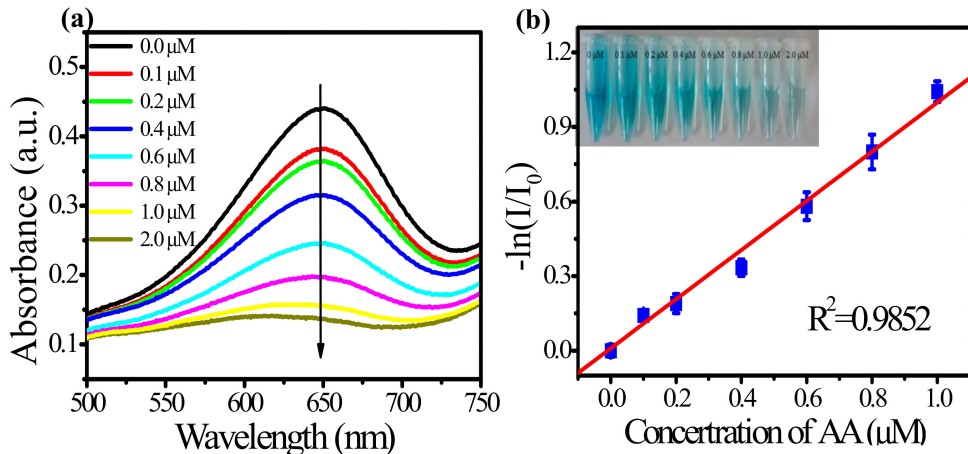

**Figure 7.** (**a**) Effect of AA on the absorbance spectra of the TMB–Au-tipped Pt NRs–$H_2O_2$–AA system. (**b**) Linear calibration plots of $A_0A^{-1}$ against AA concentrations. (Inset): Color changes of solutions with different concentrations of AA.

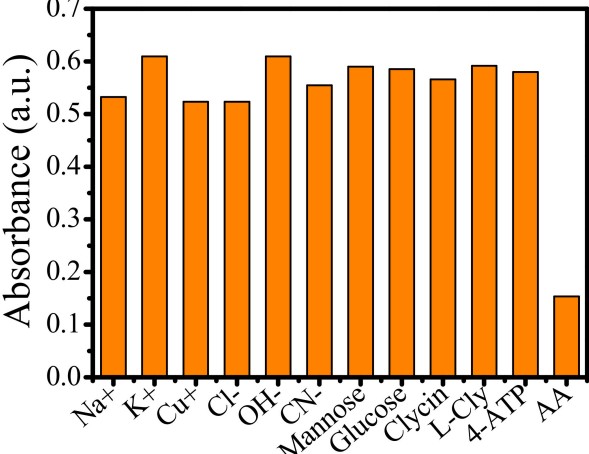

**Figure 8.** Absorbance intensity of the TMB–Au-tipped Pt NRs–$H_2O_2$ system in response to AA and interferential substances. The concentration of AA and interferential substances was 1.0 μM.

As an antioxidant, AA participates in many physiological, biochemical processes. Insufficient or excessive AA can lead to anemia and scurvy. Therefore, it is necessary to develop an effective method with high selectivity for the determination of AA in serum. The interference in serum mainly include ions and amino acid. To investigate the specificity of Au-tipped Pt NRs for AA, the absorption intensity of 650 nm at different interferences was studied, as shown in Figure 8. The detection system was kept at a total volume of 1.5 mL. The AA and common substances, including ions, mannose, glucose, cyclin, L-cly and 4-ATP, were investigated to confirm the effect of the colorimetric assay. Then, 0.8 mL TMB and 0.2 mL Au-tipped Pt NRs were mixed with 0.3 mL $H_2O_2$. The above substances were added to the above reaction solution and kept the solution at room temperature for 5 min. The results show that interferences had no obvious influence on the colorimetric assay. AA may inhibit the colorimetric process. The results demonstrated that the colorimetric reaction has high specificity for AA based on Au-tipped Pt NRs.

## 3. Materials and Methods

### 3.1. Materials

Sodium borohydride ($NaBH_4$), cetyltrimethylammonium bromide (CTAB), chloroauric acid ($HAuCl_4 \cdot 3H_2O$), L-ascorbic acid (AA), potassium tetrachloroplatinate (II) ($K_2PtCl_4$), aqueous solution (HCl) (28%) were purchased from Sinopharm Chemical Reagent Company, Ltd. (Beijing, China). 3,3',5,5'-tetramethylbenzidine (TMB) was purchased from Sigma-Aldrich (St. Louis, MO, USA). Silver nitrate ($AgNO_3$) and other metal salts were obtained from Aladdin Regent Company, Ltd. (Shanghai, China). All the chemicals were used without further purification, and all aqueous solutions were prepared with ultrapure water.

### 3.2. Preparation of Au NRs

Au NRs were synthesized using a seed-mediated growth procedure. First, CTAB (9.75 mL, 0.1 M) was mixed with $HAuCl_4$ (0.25 mL, 0.01 M). Then, ice-cold $NaBH_4$ (0.6 mL, 0.01 M) was added while stirring magnetically. After 3 min, stirring was stopped, and the color changed from bright yellow to brown, indicating the formation of Au seed. Before further use, the seed solution was left undisturbed at room temperature (RT) for 2–5 h. The growth solution of Au NRs consisted of $HAuCl_4$ (2 mL, 0.01 M), $AgNO_3$ (0.4 mL, 0.01 M), CTAB (40 mL, 0.1 M), AA (0.32 mL, 0.1 M) and HCl (0.8 mL, 1 M). Then, the growth solution was inoculated with 0.1 mL seed solution to initiate the growth of Au NRs. After 12 h, the reaction was stopped, and the collected solution was stored at 0 °C. The concentration of Au NRs was 2.26 nM.

### 3.3. Preparation of Au-tipped Pt NRs

Each of the six Au NR samples (8 mL), without purified solutions, was mixed with corresponding concentration (0.5, 1.0, 3.0, 5.0, 10, 50 mM) of 1 mL $H_2PtCl_4$ aqueous solutions. Then, different amounts of AA were added to maintain the $AA/H_2PtCl_4$ molar ratio of 10. The mixture was shaken vigorously for about 10 s and incubated in a water bath at 30 °C for 12 h. Lastly, the collected solution was stored at 0 °C, and the final concentration of Au-tipped Pt NRs was 2.26 nM.

### 3.4. Colorimetric Detection of AA

First, 2.26 nM Au-tipped Pt NRs (0.2 mL) and TMB (0.8 mL) were mixed. Then 0.3 mL $H_2O_2$ (6 μM) and 0.2 mL AA (0, 0.2, 0.4, 0.6, 0.8 or 1.0 μM) were added to the reaction mixtures which were incubated at RT for 5 min. Finally, the absorbance was recorded at 652 nm using a spectrophotometer (PTI-Quanta Master 400, Tucson in USA).

### 3.5. Validation of the Colorimetric Method for the Detection of AA

The water solution consisting of various ions or compounds was diluted to the equivalent concentration (6 μM) of the detected AA. Then, the reaction solution was added to 0.2 mL of Au-tipped Pt NRs, 0.8 mL TMB and 0.3 mL of $H_2O_2$ and incubated at RT for 2 min. Subsequently, 0.2 mL of interfering substance was added, and the absorption was recorded by a spectrophotometer.

## 4. Conclusions

In this paper, platinum and platinum nanocomposites were produced by laser-induced method. The Pt nanoparticles were grown at the tip of Au NRs and analyzed by TEM. Next, using the surface plasma absorption spectra and XRD, we found that the longitudinal plasma resonance absorption peak of the material showed a red-shift and the high crystalline energy surface of the pure Au nanorod was covered by platinum nanoparticles. Moreover, catalytic properties of the materials were tested, and detection for AA was performed using the same. We suggest that these findings potentially provide new avenues for the detection of biomolecules with naked eyes, great ease and portability.

**Author Contributions:** Conceived and designed the experiments: H.G. and L.W.; H.G., W.H., J.L. performed the experiments: (Sections 2.1 and 2.2); H.G., W.H., J.Q., H.L. (Sections 2.3 and 2.4) and H.G., L.W. analyzed the data: (Sections 2.1–2.4). H.G., J.L., J.Q. contributed reagents/materials/analysis. H.G., W.H. contributed to the writing of the manuscript. L.W., H.L. revised the paper. All authors have read and agreed to the published version of the manuscript.

**Funding:** This work was supported by the National Natural Science Foundation of China (No.31401086) and the Science and Technology Development Program of Jilin Province, China (No.20200404114YY).

**Conflicts of Interest:** There are no conflicts to declare.

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
