# Peer review of "The Mimic Enzyme Properties of Au@PtNRs and the Detection for Ascorbic Acid Based on Their Catalytic Properties"

_catalysts, doi:10.3390/catal10111285_

Round 1
Reviewer 1 Report
The paper deals with the synthesis and use of Au-tipped-Pt nanorods (NRs) for an application which takes advantage of their peroxidase-like activity. In particular, a specific use for ascorbic acid (AA) presence detection is suggested which could be revealed via spectrophotometric monitoring the decreased oxidation of a chromogenic substrate 3, 3’, 5, 5’-tetramethylbenzidine (TMB) in the presence of H2O2. The paper is interesting and could be published in the journal, also as an example of nanoenzymes application for biosensors, provided that some key points are addressed.
- The performance comparison of simple Au nanorods (without Pt), with respect to peroxidase-like activity, should be at least declared if not carefully evaluated and discussed.
- If the suggested method should be considered specific for AA detection other antioxidants should be tested as possible interfering agents. Otherwise, the possible similar effect originated by the presence of such species acting similarly to AA should be discussed.
Minor point
- The “average length diameter ratio of 3.3.” (length to diameter, L/D ?) page 3 line 91 Please clarify.
Author Response
Dear editor and reviewers:
Thank you very much for your letter. We are pleased to know that our work was rated as potentially acceptable for publication in Catalysts, subject to adequate revision. We thank the reviewers for the time and effort that they have put into reviewing the previous version of the manuscript. Their suggestions have enabled us to improve our work. Based on the suggestion provided, we upload the file of the revised manuscript. The modified parts of the paper were marked in red.
The point-by-point response to the comments raised by the reviewers is appended to this letter. Our responses are given in blue.
We would also like to thank you for allowing us to resubmit a revised copy of the manuscript.
Sincerely
Liping Wang Ph.D.

Reviewer 2 Report
This manuscript describes the colorimetric detection of ascorbic acid, which is based on the peroxidase-activity of Au@Pt nanorods. In the manuscript, the molecular oxygen (O2) directly oxidizes TMB, which is inhibited by the presence of ascorbic acid. This means that O2 is an oxidant of TMB. However, the oxidation of TMB (or ascorbic acid) by molecular O2 is very slow unless the reaction is conducted under high pressure of O2. The reviewer thinks that the tiny produced O2 from H2O2 is not the main oxidant of these reactants. More plausibly, OH radical should be the main oxidant. Furthermore, there was no description that experimentally confirms the O2 production on the surface of Au@Pt nanorods. From the mechanistic viewpoint, this manuscript should be revised. The mechanism the authors present is too speculative.
Author Response

(The authors gave the same response as above.)

Round 2
Reviewer 1 Report
Althoug (in part) answered to the reviewer in the author's reply, none of the suggested statement/revision/discussion related to questions 1-3 has been included in the revised text. Please report or discuss the points in the text.
In particular answer to question 2
"Antioxidants are a class of biological substances, each of which has a special function. They don't interact with each other. So we just need to determine the function and properties of a substance."
extends the finding reported in Fig. 8 well over the possibly interfering substances there considered, therefore the author's reply sentence should be not only reported but also referenced.
Author Response
Dear editor and reviewers:
Thank you very much for your letter. We are pleased to know that our work was rated as potentially acceptable for publication in Catalysts, subject to adequate revision. We thank the reviewers for the time and effort that they have put into reviewing the previous version of the manuscript. Their suggestions have enabled us to improve our work. Based on the suggestion provided, we upload the file of the revised manuscript. The modified parts of the paper were marked in red.The point-by-point response to the comments raised by the reviewers is appended to this letter. Our responses are given in blue.
We would also like to thank you for allowing us to resubmit a revised copy of the manuscript.
Sincerely,
Liping Wang
School of Life Sciences, Jilin University, No.2699 Qianjin Street, Changchun City
130012, People’s Republlic of China.
E-mail address: wanglp@jlu.edu.cn (Liping Wang).
To reviewer:
1.The performance comparison of simple Au nanorods (without Pt), with respect to peroxidase-like activity, should be at least declared if not carefully evaluated and discussed.
Answer: Au NRs, Pt NPs and Au@Pt exhibit catalytic properties. The spin trapping technique were used to confirm the catalytic activity of Au NRs, Pt NPs and Au@Pt NRs as shown in Fig. 3 (a). The results show that hydroxyl radical is formed. However, their catalytic properties indicate that Au@Pt NRs are highest among these materials. So, Au@Pt NRs were confirmed to own obvious high peroxidase-like activity.
- If the suggested method should be considered specific for AA detection other antioxidants should be tested as possible interfering agents. Otherwise, the possible similar effect originated by the presence of such species acting similarly to AA should be discussed.
Answer: AA as antioxidant participates in many physiological biochemical process. Insufficient or excessive AA can lead to scurvy and anemia. Therefore, it is import to develop a convenient and rapid method with high selectivity for the determination of AA in serum. The interference in serum mainly include ions and amino acid. So, the selective experiment for AA was performed based on ions and amino acid as shown in Fig.8.
3.The “average length diameter ratio of 3.3.” (length to diameter, L/D ?) page 3 line 91 Please clarify.
Answer: For Au NRs, two absorption bands including longitudinal localized surface plasmon resonance (LSPR) at 724 nm and transverse localized surface plasmon resonance (TSPR) at 510 nm can be seen in UV-vis absorption spectrum. They are corresponding to the electron oscillations along the long axis and short axis of Au NRs, respectively. L/D means the length-to-width ratio of Au NRs. The increase of L/D may induces the red-shift of LSPR peak. Herein, the L/D of Au NRs remains the same during the preparation of Au-tipped-Pt NRs.
Reviewer 2 Report
The authors detected OH radical formation by ESR, supporting the proposed reaction mechanism. I recommend this manuscript for publication.
Author Response
Dear editor and reviewers:
Thank you very much for your letter. We are pleased to know that our work was rated as potentially acceptable for publication in Catalysts, subject to adequate revision. We thank the reviewers for the time and effort that they have put into reviewing the previous version of the manuscript. Their suggestions have enabled us to improve our work. Based on the suggestion provided, we upload the file of the revised manuscript. The modified parts of the paper were marked in red.
We would also like to thank you for allowing us to resubmit a revised copy of the manuscript.
Sincerely,
Liping Wang
School of Life Sciences, Jilin University, No.2699 Qianjin Street, Changchun City
130012, People’s Republlic of China.
E-mail address: wanglp@jlu.edu.cn (Liping Wang).
Round 3
Reviewer 1 Report
The Authors have essentially dealt with the reviewer questions and revised the manuscript accordingly.